# Arbitrary Conditional Distributions with Energy

**Ryan R. Strauss**
Department of Computer Science
UNC at Chapel Hill
Chapel Hill, NC 27514
rrs@cs.unc.edu

**Junier B. Oliva**
Department of Computer Science
UNC at Chapel Hill
Chapel Hill, NC 27514
joliva@cs.unc.edu

## Abstract

Modeling distributions of covariates, or *density estimation*, is a core challenge in unsupervised learning. However, the majority of work only considers the *joint* distribution, which has limited utility in practical situations. A more general and useful problem is *arbitrary conditional density estimation*, which aims to model *any* possible conditional distribution over a set of covariates, reflecting the more realistic setting of inference based on prior knowledge. We propose a novel method, Arbitrary Conditioning with Energy (ACE), that can simultaneously estimate the distribution $p(\mathbf{x}_u \mid \mathbf{x}_o)$ for all possible subsets of unobserved features $\mathbf{x}_u$ and observed features $\mathbf{x}_o$. ACE is designed to avoid unnecessary bias and complexity — we specify densities with a highly expressive energy function and reduce the problem to only learning one-dimensional conditionals (from which more complex distributions can be recovered during inference). This results in an approach that is both simpler and higher-performing than prior methods. We show that ACE achieves state-of-the-art for arbitrary conditional likelihood estimation and data imputation on standard benchmarks.

## 1 Introduction

Density estimation, a core challenge in machine learning, attempts to learn the probability density of some random variables given samples from their true distribution. The vast majority of work on density estimation focuses on the *joint* distribution $p(\mathbf{x})$ [10, 4, 25, 11, 23, 22, 6], i.e., the distribution of all variables taken together. While the joint distribution can be useful (e.g., to learn the distribution of pixel configurations that represent human faces), it is limited in the types of predictions it can make. We are often more interested in *conditional* probabilities, which communicate the likelihood of an event *given* some prior information. For example, given a patient's medical history and symptoms, a doctor determines the likelihoods of different illnesses and other patient attributes. Conditional distributions are often more practical since real-world decisions are nearly always informed by prior information.

However, we often do not know ahead of time *which* conditional distribution (or distributions) will be of interest. That is, we may not know which features will be known (observed) or which will be inferred (unobserved). For example, not every patient that a doctor sees will have had the same tests performed (i.e., have the same observed features). A naïve approach requires building an exponential number of models to cover all possible cases (one for every conditional distribution), which quickly becomes intractable. Thus, an intelligent system needs to understand the intricate conditional dependencies between all arbitrary subsets of covariates, and it must do so with a single model to be practical.

In this work, we consider the problem of learning the conditional distribution $p(\mathbf{x}_u \mid \mathbf{x}_o)$ for any arbitrary subsets of unobserved variables $\mathbf{x}_u \in \mathbb{R}^{|u|}$ and observed variables $\mathbf{x}_o \in \mathbb{R}^{|o|}$, where $u, o \subseteq \{1, \dots, d\}$ and $o \cap u = \emptyset$. We propose a method, Arbitrary Conditioning with Energy (ACE),

35th Conference on Neural Information Processing Systems (NeurIPS 2021).

that can assess any conditional distribution over any subset of random variables, using a single model. ACE is developed with an eye for simplicity — we reduce the arbitrary conditioning problem to the estimation of one-dimensional conditional densities (with arbitrary observations), and we represent densities with an energy function, which fully specifies unnormalized distributions and has the freedom to be instantiated as any arbitrary neural network.

We evaluate ACE on benchmark datasets and show that it outperforms current methods for arbitrary conditional/marginal density estimation. ACE remains effective when trained on data with missing values, making it applicable to real-world datasets that are often incomplete, and we find that ACE scales well to high-dimensional data. Also, unlike some prior methods (e.g., [20]), ACE can naturally model data with both continuous and discrete values.

Our contributions are as follows: 1) We develop the first energy-based approach to arbitrary conditional density estimation, which eliminates restrictive biases (e.g. normalizing flows, Gaussian mixtures) imposed by common alternatives. 2) We empirically demonstrate that ACE is state-of-the-art for arbitrary conditional density estimation and data imputation. 3) We find that complicated prior approaches can be easily outperformed with a simple scheme that uses mixtures of Gaussians and fully-connected networks.

## 2   Previous Work

Several methods have been previously proposed for arbitrary conditioning. Sum-Product Networks are specially designed to only contain sum and product operations and can produce arbitrary conditional or marginal likelihoods [27, 3]. The Universal Marginalizer trains a neural network with a cross-entropy loss to approximate the marginal posterior distributions of all unobserved features conditioned on the observed ones [5]. VAEAC is an approach that extends a conditional variational autoencoder by only considering the latent codes of unobserved dimensions [14], and NeuralConditioner uses adversarial training to learn each conditional distribution [1]. DMFA uses factor analysis to have a neural network output the parameters of a conditional Gaussian density for the missing features given the observed ones [28]. The current state-of-the-art is ACFlow, which extends normalizing flow models to handle any subset of observed features [20].

Unlike VAEAC, ACE does not suffer from mode collapse or blurry samples. ACE is also able to provide likelihood estimates, unlike NeuralConditioner which only produces samples. DMFA is limited to learning multivariate Gaussians, which impose bias and are harder to model than one-dimensional conditionals. While ACFlow can analytically produce normalized likelihoods and samples, it is restricted by a requirement that its network consist of bijective transformations with tractable Jacobian determinants. Similarly, Sum-Product Networks have limited expressivity due to their constraints. ACE, on the other hand, exemplifies the appeal of energy-based methods as it has no constraints on the parameterization of the energy function.

Energy-based methods have a wide range of applications within machine learning [17], and recent work has studied their utility for density estimation. Deep energy estimator networks [30] and Autoregressive Energy Machines [22] are both energy-based models that perform density estimation. However, both of these methods are only able to estimate the joint distribution.

Much of the previous work on density estimation relies on an autoregressive decomposition of the joint density according to the chain rule of probability. Often, the model only considers a single (arbitrary) ordering of the features [22, 24]. Uria et al. [33] proposed a method for assessing joint likelihoods based on any arbitrary ordering, where they use masked network inputs to effectively share weights between a combinatorial number of models. Germain et al. [8] also consider a shared network for joint density estimation with a constrained architecture that enforces the autoregressive constraint in joint likelihoods. In this work, we construct an order-agnostic weight-sharing technique not for joint likelihoods, but for arbitrary conditioning. Moreover, we make use of our weight-sharing scheme to estimate likelihoods with an energy based approach, which avoids the limitations of the parametric families used previously (e.g., mixtures of Gaussians [33], or Bernoullis [8]). Query Training [16] is a method for answering probabilistic queries. It also takes the approach of computing one-dimensional conditional likelihoods but does not directly pursue an autoregressive extension of that ability.

The problem of imputing missing data has been well studied, and there are several approaches based on classic machine learning techniques such as $k$-nearest neighbors [32], random forests [31], and autoencoders [9]. More recent work has turned to deep generative models. GAIN is a generative adversarial network (GAN) that produces imputations with the generator and uses the discriminator to discern the imputed features [34]. Another GAN-based approach is MisGAN, which learns two generators to model the data and masks separately [19]. MIWAE adapts variational autoencoders by modifying the lower bound for missing data and produces imputations with importance sampling [21]. ACFlow can also perform imputation and is state-of-the-art for imputing data that are missing completely at random (MCAR) [20].

While it is not always the case that data are missing at random, the opposite case (i.e., missingness that depends on unobserved features' values) can be much more challenging to deal with [7]. Like many data imputation methods, we focus on the scenario where data are MCAR, that is, where the likelihood of being missing is independent of the covariates' values.

## 3   Background

### 3.1   Arbitrary Conditional Density Estimation

A probability density function (PDF), $p(\mathbf{x})$, outputs a nonnegative scalar for a given vector input $\mathbf{x} \in \mathbb{R}^d$ and satisfies $\int p(\mathbf{x}) \, d\mathbf{x} = 1$. Given a dataset $\mathcal{D} = \{\mathbf{x}^{(i)}\}_{i=1}^N$ of *i.i.d.* samples drawn from an unknown distribution $p^*(\mathbf{x})$, the object of density estimation is to find a model that best approximates the function $p^*$. Modern approaches generally rely on neural networks to directly parameterize the approximated PDF.

Arbitrary conditional density estimation is a more general task where we estimate the conditional density $p^*(\mathbf{x}_u \mid \mathbf{x}_o)$ for all possible subsets of observed features $o \subset \{1, \ldots, d\}$ (i.e., features whose values are known) and corresponding subset of unobserved features $u \subset \{1, \ldots, d\}$ such that $o$ and $u$ do not intersect. Here $\mathbf{x}_o \in \mathbb{R}^{|o|}$ and $\mathbf{x}_u \in \mathbb{R}^{|u|}$. The estimation of joint or marginal likelihoods is recovered when $o$ is the empty set. Note that marginalization to obtain $p(\mathbf{x}_o) = \int p(\mathbf{x}_u, \mathbf{x}_o) \, d\mathbf{x}_u$ (and hence $p(\mathbf{x}_u \mid \mathbf{x}_o) = \frac{p(\mathbf{x}_u, \mathbf{x}_o)}{p(\mathbf{x}_o)}$) is intractable in performant generative models like normalizing flow based models [4, 20]; thus, we propose a weight-sharing scheme below.

### 3.2   Energy-Based Models

Energy-based models capture dependencies between variables by assigning a nonnegative scalar *energy* to a given arrangement of those variables, where energies closer to zero indicate more desirable configurations [17]. Learning consists of finding an energy function that outputs low energies for correct values. We can frame density estimation as an energy-based problem by writing likelihoods as a Boltzmann distribution

$$p(\mathbf{x}) = \frac{e^{-\mathcal{E}(\mathbf{x})}}{Z}, \qquad Z = \int e^{-\mathcal{E}(\mathbf{x})} \, d\mathbf{x} \tag{1}$$

where $\mathcal{E}$ is the energy function, $e^{-\mathcal{E}(\mathbf{x})}$ is the unnormalized likelihood, and $Z$ is the normalizer.

Energy-based models are appealing due to their relative simplicity and high flexibility in the choice of representation for the energy function. This is in contrast to other common approaches to density estimation such as normalizing flows [4, 20], which require invertible transformations with Jacobian determinants that can be computed efficiently. Energy functions are also naturally capable of representing non-smooth distributions with low-density regions or discontinuities.

## 4   Arbitrary Conditioning with Energy

We are interested in approximating the probability density $p(\mathbf{x}_u \mid \mathbf{x}_o)$ for any arbitrary sets of unobserved features $\mathbf{x}_u$ and observed features $\mathbf{x}_o$. We approach this by decomposing likelihoods into products of one-dimensional conditionals, which makes the learned distributions much simpler. This concept is a basic application of the chain rule of probability, but has yet to be thoroughly exploited for arbitrary conditioning. During training, ACE estimates distributions of the form $p(x_{u'_i} \mid \mathbf{x}_{o'})$,

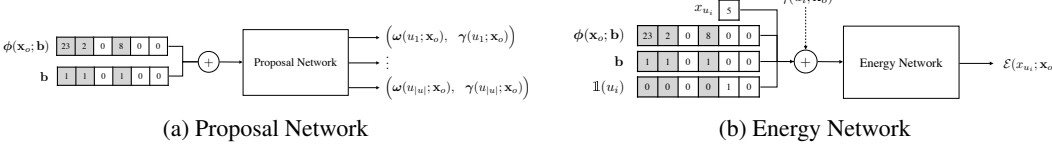

(a) Proposal Network       (b) Energy Network

Figure 1: Overview of the networks used in ACE. The plus symbol refers to concatenation.

where $x_{u'_i}$ is a scalar. During inference, more complex distributions can then be recovered with an autoregressive decomposition: $p(\mathbf{x}_u \mid \mathbf{x}_o) = \prod_{i=1}^{|u|} p(x_{u'_i} \mid \mathbf{x}_{o \cup u'_{<i}})$, where $u'$ is an arbitrary permutation of $u$ and $u'_{<i} = \{u'_1, \ldots, u'_{i-1}\}$. This approach is appealing because the estimated distributions are over a one-dimensional domain, allowing one to use a myriad of flexible estimators.

We adopt an energy-based approach (similar to AEMs [22]), which affords a large degree of flexibility in modeling the exponentially many conditional distributions at hand — we are free to represent the energy function with an arbitrary, and highly expressive, neural network that directly outputs unnormalized likelihoods. This contrasts with the current state-of-the-art for arbitrary conditional density estimation, which is limited by normalizing flow transformations [20]. Energy functions are highly expressive and naturally model complex (i.e., multimodal, non-smooth) densities, as they avoid a parametric-family on the shape of the distribution. Our main contribution is a method, Arbitrary Conditioning with Energy (ACE), for computing arbitrary conditional likelihoods with energies, one dimension at a time.[1]

### 4.1 Decomposing Densities

We decompose the arbitrary conditioning task into $1d$-domain arbitrary conditional estimation problems. This follows from the chain rule of probability, which allows us to write

$$p(\mathbf{x}_u \mid \mathbf{x}_o) = \prod_{i=1}^{|u|} p\left(x_{u'_i} \mid \mathbf{x}_{o \cup u'_{<i}}\right), \tag{2}$$

where $u'$ is an arbitrary permutation of $u$, $x_{u'_i}$ is the $i^{\text{th}}$ unobserved feature given by $u'$, and $u'_{<i} = \{u'_1, \ldots, u'_{i-1}\}$. The one-dimensional conditionals in Equation 2 are themselves arbitrary conditionals due to the choice of permutation. Thus, the arbitrary conditioning problem may be reduced to one-dimensional arbitrary conditional estimation. If we can compute any likelihood of the form $p(x_i \mid \mathbf{x}_{o'})$ for $o' \subset \{1, \ldots, d\}$, we can compute any distribution over the features. This includes all possible conditional distributions, marginal distributions, and the joint distribution. Although the reduction of arbitrary conditioning to $1d$ conditional estimation is seemingly straightforward, this is, to the best of our knowledge, the first work to leverage this formulation.

We train our model non-autoregressively, to output the likelihood $p(x_i \mid \mathbf{x}_{o'})$ for any $x_i$ and $\mathbf{x}_{o'}$, where $i \in \{1, \ldots, d\} \setminus o'$. During inference, an autoregressive procedure that repeatedly applies the chain rule can be used to compute likelihoods of the form $p(\mathbf{x}_u \mid \mathbf{x}_o)$. In practice, this consists of simply adding the previously considered unobserved dimension to the observed set before moving to the next step in the chain.[2]

### 4.2 Likelihoods from Energies

We express likelihoods in terms of energies by modifying Equation 1 to write

$$p(x_{u_i} \mid \mathbf{x}_o) = \frac{e^{-\mathcal{E}(x_{u_i}; \mathbf{x}_o)}}{Z_{u_i; \mathbf{x}_o}}, \tag{3}$$

and we choose to represent the energy function as a neural network. In order to avoid learning a different network for each conditional, we adopt a weight-sharing scheme (e.g. [33, 14, 20]). That is,

---

[1]An implementation of ACE is available at `https://github.com/lupalab/ace`.

[2]One could also implement this with a masked network architecture (e.g., MADE [8]) that can more efficiently compute autoregressive likelihoods. We choose not to do this for simplicity and to enable the use of arbitrary, and more expressive, architectures.

we use a bitmask $\mathbf{b} \in \{0, 1\}^d$ that indicates which features are observed and define a zero-imputing function $\phi(\mathbf{x}_o; \mathbf{b})$ that returns a $d$-dimensional vector where unobserved features are replaced with zeros (see Figure 5 in Appendix). These are then used as inputs to the energy network (see Figure 1b).

**Approximating normalized likelihoods.** We can use Equation 3 to compute normalized likelihoods, but only if the normalizing constant $Z_{u_i;\mathbf{x}_o}$ is known. Directly computing the normalizer is intractable in general. However, importance sampling can be used to obtain an estimate [2]. Nash and Durkan [22] show that this technique is sufficiently accurate when considering one-dimensional conditionals for joint density estimation, and we adapt the approach for arbitrary conditioning. Assuming access to a proposal distribution $q(x_{u_i} \mid \mathbf{x}_o)$ which is similar to the target distribution, we approximate $Z_{u_i;\mathbf{x}_o}$ as

$$Z_{u_i;\mathbf{x}_o} = \int e^{-\mathcal{E}(x_{u_i};\mathbf{x}_o)} \, \mathrm{d}x_{u_i} = \int \frac{e^{-\mathcal{E}(x_{u_i};\mathbf{x}_o)}}{q(x_{u_i} \mid \mathbf{x}_o)} q(x_{u_i} \mid \mathbf{x}_o) \, \mathrm{d}x_{u_i} \tag{4}$$

$$\approx \frac{1}{S} \sum_{s=1}^{S} \frac{e^{-\mathcal{E}(x_{u_i}^{(s)};\mathbf{x}_o)}}{q(x_{u_i}^{(s)} \mid \mathbf{x}_o)}, \quad x_{u_i}^{(s)} \sim q(x_{u_i} \mid \mathbf{x}_o) \,. \tag{5}$$

For some problems, we may have access to a good proposal distribution ahead of time, but in general, we can learn one in parallel with the energy network. We note that importance sampling is also more computationally efficient than the simpler alternative of using points from a grid. Since distributions may be non-smooth and span a large domain, fewer samples from a good proposal distribution will be able to obtain as good of an estimate as a grid that has many more points, which in turn will require fewer evaluations of the energy network. Furthermore, generating samples from the proposal is very efficient, as it only requires one neural network evaluation to obtain the parameters of a tractable parametric distribution, which can then be easily sampled.

### 4.3 Training ACE

We learn the proposal distribution alongside the energy function by having a neural network output the parameters of a tractable parametric distribution. For the proposal network, we again share weights between all conditionals as is done with the energy network. The proposal network accepts a concatenation of $\mathbf{b}$ and $\phi(\mathbf{x}_o; \mathbf{b})$ as input, and it outputs the parameters $\boldsymbol{\omega}(u_i; \mathbf{x}_o)$ for a mixture of Gaussians for each unobserved dimension $u_i$. The proposal network also outputs a latent vector, $\boldsymbol{\gamma}(u_i; \mathbf{x}_o)$, for each unobserved dimension, which is used as input to the energy network in order to enable weight sharing between the proposal and energy networks [22]. Using a different latent vector for each feature allows the latent vectors to represent the fact that different unobserved features may depend on $\mathbf{x}_o$ in different ways.

We then estimate the normalizing constants in Equation 3 using importance sampling:

$$\hat{Z}_{u_i;\mathbf{x}_o} = \frac{1}{S} \sum_{s=1}^{S} \frac{e^{-\mathcal{E}(x_{u_i}^{(s)};\mathbf{x}_o;\boldsymbol{\gamma}(u_i;\mathbf{x}_o))}}{q(x_{u_i}^{(s)} \mid \boldsymbol{\omega}(u_i;\mathbf{x}_o))} \tag{6}$$

where $x_{u_i}^{(s)}$ is sampled from $q(x_{u_i} \mid \boldsymbol{\omega}(u_i; \mathbf{x}_o))$. This in turn leads to the following approximation of the log-likelihood of $x_{u_i}$ given $\mathbf{x}_o$:

$$\log p(x_{u_i} \mid \mathbf{x}_o) \approx -\mathcal{E}(x_{u_i}; \mathbf{x}_o; \boldsymbol{\gamma}(u_i; \mathbf{x}_o)) - \log \hat{Z}_{u_i;\mathbf{x}_o}, \tag{7}$$

where we use abbreviated notation in the previous two equations and omit the bitmask $\mathbf{b}$ for greater readability. Refer to Figure 1 for the precise inputs of each network.[3]

Since Equation 2 gives us a way to autoregressively compute $p(\mathbf{x}_u \mid \mathbf{x}_o)$ as a chain of one-dimensional conditionals, we only concern ourselves with learning $p(x_{u_i} \mid \mathbf{x}_o)$ for arbitrary $u_i$ and $\mathbf{x}_o$. Thus, for a given data point $\mathbf{x}$, we randomly partition it into $\mathbf{x}_o$ and $\mathbf{x}_u$ and jointly optimize the proposal and energy networks with the maximum-likelihood objective

$$J(\mathbf{x}_o; \mathbf{x}_u; \boldsymbol{\theta}) = \sum_{i=1}^{|u|} \log p(x_{u_i} \mid \mathbf{x}_o) + \sum_{i=1}^{|u|} \log q(x_{u_i} \mid \mathbf{x}_o), \tag{8}$$

---

[3]It is not strictly necessary for the energy network to include $\mathbf{x}_o$ as input, since the latent vectors can encode that information. However, doing so acts like a sort of skip-connection, which often helps with deeper networks.

where $\boldsymbol{\theta}$ holds the parameters of both the energy and proposal networks. Because we want to optimize the proposal and energy distributions independently, gradients are stopped on proposal samples and proposal likelihood before they are used in Equation 6. We note that optimizing Equation 8 can be interpreted as a stochastic approximation to optimizing the full autoregressive likelihoods $p(\mathbf{x}_u \mid \mathbf{x}_o)$ (i.e., Equation 2). For a given data point, we are choosing to optimize individual $1d$-conditionals from numerous hypothetical autoregressive chains, rather than all of the $1d$-conditionals from a single autoregressive chain. That is, each random set of observed values that is encountered during training (in Equation 8) can be seen as a single factor in the product in Equation 2 for some $p(\mathbf{x}_u \mid \mathbf{x}_o)$. Over the course of training though, the same conditionals are ultimately being evaluated in either case.

The negative of Equation 8 is minimized with Adam [15] over a set of training data, where observed and unobserved sets are selected at random for each minibatch (see Section 5). In some cases, we found it useful to include a regularization term in the loss that penalizes the energy distribution for large deviations from the proposal distribution. We use the mean-square error (MSE) between the proposal likelihoods and energy likelihoods as a penalty, with gradients stopped on the proposal likelihoods in the error calculation. The coefficient of this term in the loss is a hyperparameter.

### 4.4 Inference

#### 4.4.1 Likelihoods

Recall that our model learns one-dimensional conditionals: $p(x_{u_i} \mid \mathbf{x}_o)$. Thus, to obtain a complete likelihood for $\mathbf{x}_u$, we employ an autoregressive application of the chain rule (see Equation 2). The pseudocode for this procedure is presented in the Appendix (see Algorithm 1). Since the values of $\mathbf{x}_u$ are known ahead of time, each one-dimensional conditional can be evaluated in parallel as a batch, allowing the likelihood $p(\mathbf{x}_u \mid \mathbf{x}_o)$ to be computed efficiently.

Importantly, the order in which each unobserved dimension is evaluated does not matter. As argued in previous work [33, 8], this can be considered advantageous, since we can effectively leverage multiple orderings at test time to obtain an ensemble of models. However, this does incur extra computational cost during inference. We note that a model which perfectly captures the true densities would give consistent likelihoods for all possible orderings (thus evaluating only one ordering would suffice). However, current order-agnostic methods (such as ACE or [33, 8]) do not inherently satisfy this constraint. In the Appendix, we study how these inconsistencies can be mitigated in ACE, but we leave a true solution to this challenge for future work.

#### 4.4.2 Imputing

Sampling allows us to obtain multiple possible values for the unobserved features that are diverse and realistic.[4] However, these are not always the primary goals. For example, in the case of data imputation, we may only want a single imputation that aims to minimize some measure of error (see Section 5.1.2). Thus, rather than imputing true samples, we might prefer to impute the mean of the learned distribution. In this case, we forego autoregression and directly obtain the mean of each distribution $p(x_{u_i} \mid \mathbf{x}_o)$ with a single forward pass. Analytically computing the mean of the proposal distribution is straightforward since we are working with a mixture of Gaussians. We estimate the mean of the energy distribution via importance sampling:

$$\mathbb{E}\left[x_{u_i}\right] \approx \sum_{s=1}^{S} \frac{r_s}{\sum_j r_j} x_{u_i}^{(s)}, \qquad r_s = \frac{p(x_{u_i} \mid \mathbf{x}_o)}{q(x_{u_i} \mid \mathbf{x}_o)} \tag{9}$$

where $x_{u_i}^{(s)}$ is sampled from $q(x_{u_i} \mid \mathbf{x}_o)$. It is worth noting that imputing the mean ignores dependencies between features in the unobserved set, so for some applications, other methods of imputing (such as multiple imputation by drawing samples) may make more sense.

#### 4.4.3 Heterogenous Data

Prior approaches to arbitrary conditioning have to make restrictive assumptions when modeling arbitrary dependencies between continuous and discrete covariates. VAEAC [14], for example, makes an assumption of conditional independence given a latent code. ACFlow [20] is *not* directly

---

[4]We describe how to generate samples from ACE in the Appendix.

Table 1: Test arbitrary conditional log-likelihoods (in nats) for UCI datasets. Higher is better. Likelihood estimates are computed with 20,000 importance samples for POWER, GAS, and HEPMASS, 10,000 importance samples for MINIBOONE, and 3,000 importance samples for BSDS. Results for ACFlow and VAEAC are taken from Li et al. [20]. The best model for each dataset and missing rate is shown in bold. Results are averaged over 5 observed masks.

| | POWER | | | GAS | | | HEPMASS | | | MINIBOONE | | | BSDS | | |
|---|---|---|---|---|---|---|---|---|---|---|---|---|---|---|---|
| Missing Rate | 0.0 | 0.1 | 0.5 | 0.0 | 0.1 | 0.5 | 0.0 | 0.1 | 0.5 | 0.0 | 0.1 | 0.5 | 0.0 | 0.1 | 0.5 |
| ACE | **0.631** | **0.633** | **0.600** | **9.643** | **9.526** | **8.530** | **-3.859** | **-4.255** | **-8.133** | **0.310** | **-0.688** | **-5.701** | **86.701** | **86.130** | **80.613** |
| ACE Proposal | 0.583 | 0.573 | 0.542 | 9.484 | 9.348 | 8.183 | -4.417 | -4.796 | -8.497 | -0.241 | -1.328 | -9.169 | 85.228 | 84.204 | 75.767 |
| ACFlow | 0.561 | 0.557 | 0.458 | 8.086 | 7.568 | 5.405 | -8.197 | -7.784 | -10.538 | -0.972 | -5.150 | -9.892 | 81.827 | 80.783 | 75.050 |
| ACFlow+BG | 0.528 | 0.510 | 0.417 | 7.593 | 7.212 | 4.818 | -6.833 | -9.670 | -10.975 | -1.098 | -3.577 | -10.849 | 81.399 | 79.745 | 73.061 |
| VAEAC | -0.042 | -0.103 | -0.343 | 2.418 | 2.823 | 1.952 | -10.082 | -10.389 | -11.415 | -3.452 | -4.242 | -9.051 | 74.850 | 74.313 | 66.628 |

applicable to discrete data given its use of the change of variable theorem. On the other hand, ACE can naturally model arbitrary dependencies between continuous and discrete covariates without any assumptions. In this setting, the proposal network outputs categorical logits for discrete features, as opposed to Gaussian mixture parameters. These logits can themselves be interpreted as energies, and we don't need to learn an additional energy function for the discrete features. Similarly, ACE could be applied to multimodal data in a straightforward fashion by conditioning the proposal distributions and energy function on a fused multimodal latent representation (which could be freely learned by arbitrary neural networks).

## 5 Experiments

### 5.1 Real-valued UCI Data

We first evaluate ACE on real-valued tabular data. Specifically, we consider the benchmark UCI repository datasets described by Papamakarios et al. [25] (see Table 6 in the Appendix).

Unlike other approaches to density estimation that require particular network architectures [8, 4, 22, 20], ACE has no such restrictions. Thus, we use a simple fully-connected network with residual connections [12, 13] for both the energy network and proposal network. This architecture is highly expressive, yet simple, and helps avoid adding unnecessary complexity to ACE. The bitmask $\mathbf{b}$, which indicates observed features, is sampled for each training example by first drawing $k \sim \mathcal{U}\{0, d-1\}$, then choosing $k$ distinct features with uniform probability to be observed. Full experimental details and hyperparameters can be found in the Appendix.

We also consider the scenario in which data features are completely missing, i.e., some features are deemed unavailable for particular instances during training and are never part of the observed or unobserved set.[5] This allows us to examine the effectiveness of ACE on incomplete datasets, which are common when working with real-world data. When training models with missing data, we simply modify the sets of observed and unobserved indices to remove any indices which have been declared missing. This is a trivial modification and requires no other change to the design or training procedure of ACE. We consider two scenarios where data are missing completely at random at a 10% and 50% rate.

### 5.1.1 Likelihood Evaluation

Table 1 presents the average arbitrary conditional log-likelihoods on held-out test data from models trained with different levels of missing data. During inference, no data is missing and $\mathbf{b}$ is drawn from a Bernoulli distribution with $p = 0.5$. Likelihoods are calculated autoregressively as described in Section 4.4.1. The order in which the unobserved one-dimensional conditionals are computed is randomly selected for each instance.

We can draw two key findings from Table 1. First, we see that our proposal distribution outperforms ACFlow in all cases. Even this exceptionally simple approach (just a fully-connected network that produces a mixture of Gaussians) can give rise to extremely competitive performance, and we see there are advantages to using decomposed densities and unrestricted network architectures. Second,

---

[5]Features are missing at the per-instance level. For example, this does not mean that the $i^{\text{th}}$ feature is never observed for all training instances.

Table 2: Test marginal log-likelihoods (in nats) for UCI datasets. Higher is better. We evaluate the marginal distributions of the first 3, 5, and 10 dimensions of each dataset (POWER and GAS don't have 10 features, so the joint likelihood over all features is reported instead). The same number of importance samples are used as in Table 1. Results for ACFlow and TAN are taken from Li et al. [20]. Bold indicates where ACE outperformed ACFlow. Results are averaged over 5 observed masks.

| | POWER | | | GAS | | | HEPMASS | | | MINIBOONE | | | BSDS | | |
|---|---|---|---|---|---|---|---|---|---|---|---|---|---|---|---|
| Dimensions | 3 | 5 | 6 | 3 | 5 | 8 | 3 | 5 | 10 | 3 | 5 | 10 | 3 | 5 | 10 |
| ACE | **-0.56** | **1.42** | **0.58** | **1.31** | **4.31** | **12.20** | **-4.00** | **-5.91** | **-10.72** | **-2.13** | **-3.80** | **-7.94** | **5.10** | **9.37** | **20.31** |
| ACE Proposal | -0.58 | 1.35 | 0.49 | 1.11 | 3.98 | 11.84 | -4.01 | -5.94 | -10.82 | -2.14 | -3.79 | -7.93 | 5.06 | 9.30 | 20.17 |
| ACFlow | -0.57 | 1.34 | 0.42 | 0.78 | 3.01 | 10.13 | -4.03 | -6.19 | -11.58 | -2.76 | -5.31 | -10.36 | 5.06 | 9.26 | 19.60 |
| TAN | -0.54 | 1.40 | 0.57 | 1.22 | 4.47 | 12.09 | -4.00 | -5.92 | -10.87 | -2.13 | -3.73 | -8.13 | 5.11 | 9.43 | 20.44 |

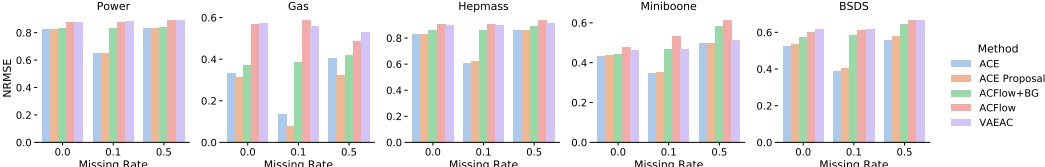

Figure 2: Normalized root-mean-square error (NRMSE) of imputations generated by ACE. Lower is better. NRMSE is computed as the root-mean-square error normalized by the standard deviation of each feature and then averaged across all features. Estimates of energy distribution means are computed with 20,000 importance samples for POWER and GAS, 10,000 importance samples for HEPMASS and MINIBOONE, and 3,000 importance samples for BSDS. Results for ACFlow and VAEAC are taken from Li et al. [20].

we find that in every case, the likelihood estimates produced by the energy function are higher than those from the proposal, illustrating the benefits of an energy-based approach which imposes no biases on the shape of the learned distributions.

We also examine the arbitrary marginal distributions learned by ACE, i.e., the unconditional distribution over a subset of features. We again test our model against ACFlow, and we additionally compare to Transformation Autoregressive Networks (TAN) [23], which are designed only for joint likelihood estimation. A separate TAN model has to be trained for each marginal distribution. While a single ACFlow model can estimate all marginal distributions, Li et al. [20] retrained models specifically for arbitrary marginal estimation. Contrarily, we used the same ACE models when evaluating arbitrary marginals as were used for arbitrary conditionals. Results are provided in Table 2. We find that ACE outperforms ACFlow in all cases and even surpasses TAN most of the time, even though ACFlow and TAN both received special training for marginal likelihood estimation and ACE did not.

### 5.1.2 Imputation

We also evaluate ACE for data imputation, where some elements are missing from a dataset completely at random and we seek to infer their values. ACE is naturally applied to this task — we consider $p(\mathbf{x}_u \mid \mathbf{x}_o)$, where $\mathbf{x}_u$ contains the missing features.

Figure 2 shows the normalized root-mean-square error (NRMSE) on held-out test data. Again, we consider models trained with three different levels of missing data. During inference, $\mathbf{b}$ is drawn from a Bernoulli distribution with $p = 0.5$ for the 0% and 50% missing rates and $p = 0.9$ for the 10% missing rate.[6] The means of the unobserved distributions are used as the imputed values (see Section 4.4.2).

As seen in Figure 2, ACE achieves a lower NRMSE score than ACFlow in all cases (exact numbers are available in the Appendix). These results further validate ACE's ability to accurately model arbitrary conditionals, leading us to again advocate for simple methods with few biases. It is also

---

[6]This means that for the 10% missing rate, we are imputing fewer values (only 10% as opposed to 50%) during inference than with the other two missing rates. This explains the lower errors for the 10% missing rate that we see in Figure 2. Even though there is missing data during training, we are estimating the values of fewer values based on a larger amount of observed data, which intuitively should result in more accurate imputations.

Table 3: Conditional log-likelihoods (LL), joint bits-per-dimension (BPD), and inpainting peak signal-to-noise-ratio (PSNR) for ACE and ACFlow on MNIST.

| Method | LL | BPD | PSNR |
|---|---|---|---|
| ACE | **1043.88** | **1.42** | **16.81** |
| ACE Proposal | 828.60 | 2.11 | 16.70 |
| ACFlow | 875.87 | 3.09 | 13.75 |

Table 4: Log-likelihood and imputation results for the UCI Adult dataset. NRMSE measures imputation performance for continuous features and accuracy is used for discrete features.

| Method | LL | NRMSE | Accuracy |
|---|---|---|---|
| ACE | **2.38** | 0.90 | **0.69** |
| ACE Proposal | 2.24 | **0.89** | **0.69** |
| VAEAC | -7.25 | 0.91 | 0.67 |

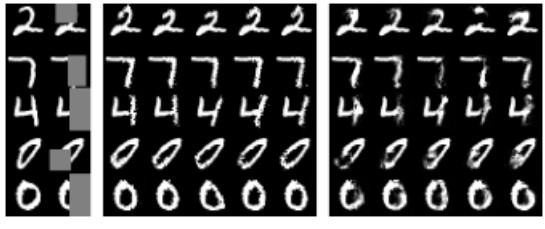

(a) Inpaintings from ACE and ACFlow. Left: Groundtruth and observed pixels. Middle: ACE samples. Right: ACFlow samples.

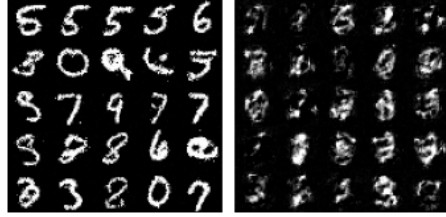

(b) Samples from the joint distribution (i.e., all pixels are unobserved). Left: ACE samples. Right: ACFlow samples.

Figure 3: MNIST samples generated by ACE and ACFlow.

worth noting that ACE and ACE Proposal do comparably in this imputation task, which estimates the first-order moment of conditional distributions. However, as evidenced in Table 1, the energy-based likelihood better captures higher-order moments.

## 5.2 High-dimensional Data

We examine ACE's ability to scale to general high-dimensional data by considering the flattened MNIST dataset [18] (i.e., each example is a 784-dimensional vector). In order to make a fair comparison, we trained a non-convolutional ACFlow model (using the authors' code) on the flattened data as well. The only change to ACE's training procedure is in the distribution of bitmasks that are sampled during training, which we detail in the Appendix.

Table 3 compares ACE and ACFlow in terms of arbitrary conditional log-likelihoods, joint bits-per-dimension (BPD), and peak signal-to-noise-ratio for inpaintings, and we see that ACE outperforms ACFlow for all three metrics. We also note that ACE's BPD of 1.42 is comparable to prior methods for joint density estimation such as TAN, MADE, and MAF, which have reported BPD of 1.19, 1.41, and 1.52 respectively (lower is better), despite the fact that these other methods can *only* model the joint distribution [23]. Qualitatively, we see that ACE produces more realistic inpaintings (see Figure 3a) and samples from the joint (see Figure 3b) than ACFlow. These findings indicate that ACE's performance scales well to high-dimensional data.

## 5.3 Mixed Continuous-Discrete Data

In order to demonstrate ACE's ability to model data with both continuous and discrete features, we conduct experiments on the UCI Adult dataset[7], which offers a relatively even balance between the number of continuous and discrete features (see below). We preprocess the data by standardizing continuous features and dropping instances with missing values. We also only keep instances for which the `native-country` feature is `United-States`.[8] The processed data has 6 continuous

---

[7]`https://archive.ics.uci.edu/ml/datasets/adult`

[8]Such instances account for about 91% of the data, and the other 9% take on any of 39 possible values. Thus we are able to discard a highly class-imbalanced feature while retaining the vast majority of the data.

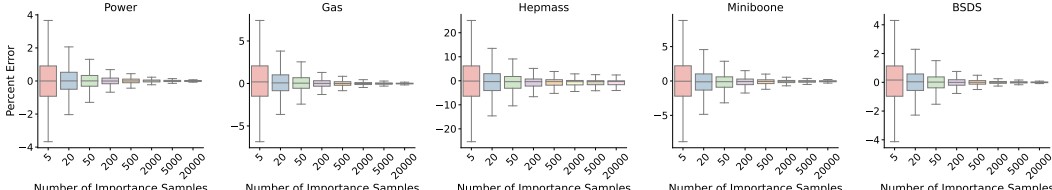

Figure 4: Percent error of normalizing constant estimates obtained with importance sampling when compared to "true" values obtained with numerical integration. Estimates grow more accurate as the number of importance samples increases. Whiskers indicate 1.5 times the IQR.

features and 8 discrete features and is split into train, validation, and test partitions of size 22003, 5501, and 13788 respectively.

We trained an ACE model and VAEAC model (using the authors' publicly available code) on this dataset and evaluated them in terms of likelihoods and imputation. For discrete features, the mode of the learned distribution is imputed (as opposed to the mean). The results are presented in Table 4, where we see that ACE outperforms VAEAC on all metrics. We also reiterate that due to its use of the change of variable theorem, ACFlow cannot be trained on this dataset.

### 5.4 Importance Sampling Accuracy

In order to better understand the impact of the number of importance samples used when estimating normalizers, we compare the importance sampling estimates from our model to "true" estimates obtained via numerical integration (similar to [22]). For each UCI dataset, we randomly select 10,000 one-dimensional conditionals (from a validation set that was not seen during training) and integrate the unnormalized energy likelihoods using a trapezoidal rule [26]. For each conditional, we then also estimate the normalizer with an increasing number of importance samples. Figure 4 shows the percentage errors of the importance sampling estimates. In most cases, a small number of samples (e.g., 20) can already produce relatively accurate estimates, and we see that the estimates grow more accurate as the number of samples increases.

## 6    Conclusion

In this work, we present a simple approach to modeling all arbitrary conditional distributions $p(\mathbf{x}_u \mid \mathbf{x}_o)$ over a set of covariates. Our method, Arbitrary Conditioning with Energy (ACE), is the first to wholly reduce arbitrary conditioning to one-dimensional conditionals with arbitrary observations and to estimate these with energy functions. By using an energy function to specify densities, ACE can more easily model highly complex distributions, and it can freely use high-capacity networks to model the exponentially many distributions at hand.

Empirically, we find that ACE achieves state-of-the-art performance for arbitrary conditional/marginal density estimation and for data imputation. For a given dataset, all of these results are produced with the same trained model. This high performance does not come at the cost of complexity — ACE is much simpler than other common approaches, which often require restrictive complexities such as normalizing flow models or networks with specially masked connections [20, 8].

Furthermore, ACE's proposal distribution, a basic mixture of Gaussians, outperforms prior methods, demonstrating that the principle of learning one-dimensional distributions is still powerful when decoupled from energy-based learning. These results emphasize that seemingly complex problems do not necessitate highly complex solutions, and we believe future work on arbitrary density estimation will benefit from similar ideas.

## Acknowledgments and Disclosure of Funding

This work was supported in part by grants NIH 1R01AA02687901A1 and NSF IIS2133595.

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
