# Appendix

## A  Sampling from ACE

Sampling the proposal distribution can be performed in an autoregressive fashion where $x_{u_i}$ is sampled from $q(x_{u_i} \mid \mathbf{x}_o)$ then added to the observed set, at which point $x_{u_{i+1}}$ can be sampled. We do this until all unobserved features have been sampled. The pseudocode for this procedure is presented in Algorithm 2.

We also want to produce samples that come from the energy function. One drawback of energy-based models is that we are unable to analytically sample the learned distribution. However, there are several methods for obtaining approximate samples. We employ a modification of the proposal sampling procedure such that many proposal samples are drawn at each step, and a single sample is then chosen from that collection based on importance weights. As the number of samples goes to infinity, this is consistent with drawing samples from the energy distribution. The pseudocode for this procedure is presented in Algorithm 3. We note that this method of sampling is closely related to sampling importance resampling [29].

## B  Algorithms

For convenience, we provide the procedure for ACE's test-time likelihood evaluation in Algorithm 1 and sampling in Algorithm 2 and Algorithm 3.

## C  Order Consistency

Because ACE can compute likelihoods by using any permutation of $u$, there are numerous ways to compute $p(\mathbf{x}_u \mid \mathbf{x}_o)$ for a given $\mathbf{x}_u$ and $\mathbf{x}_o$. However, due to inaccuracies in the learned model, we may obtain different results depending on which ordering is used. This phenomenon has surfaced in prior work as well [33, 8], where it has been argued that different orderings can be treated as an advantageous ensemble of models. This perspective is certainly useful, but ideally, our model should give equivalent likelihoods for all orderings. In order to better understand ACE's susceptibility to this problem, as well as how it may be addressed, we do a straightforward experiment in which we fine-tune trained ACE models with an additional loss term that minimizes the variance of $p(\mathbf{x}_u \mid \mathbf{x}_o)$ computed (autoregressively) over 10 permutations of $u$. Intuitively, as the expected variance over all $\mathbf{x}$ and $\mathbf{b}$ goes to zero, the distributions induced by different orderings of $u$ become the same and the model gives the same likelihood regardless of the chosen ordering.

Table 5 shows the results of these experiments. We find that ACE models can be effectively fine-tuned to produce more consistent likelihoods over different orderings, at almost no cost in performance in terms of the average likelihood. However, we see that if desired, even stronger consistency can be obtained for a slight tradeoff in the average likelihood.

Table 5: Log-likelihoods after different amounts of consistency fine-tuning. The number after the $\pm$ is the average standard deviation of $p(\mathbf{x}_u \mid \mathbf{x}_o)$ when computed over 1000 randomly chosen orderings. The coefficient refers to the weight of the variance term in the loss during fine-tuning. The 0.0 coefficient refers to the model with which the fine-tuning was initialized.

| Coefficient | 0.0 | 0.5 | 1.0 | 2.0 |
|---|---|---|---|---|
| POWER | $0.622 \pm 0.072$ | $0.622 \pm 0.063$ | $0.620 \pm 0.058$ | $0.619 \pm 0.051$ |
| GAS | $9.583 \pm 0.513$ | $9.587 \pm 0.457$ | $9.556 \pm 0.420$ | $9.497 \pm 0.376$ |
| HEPMASS | $-3.555 \pm 0.878$ | $-3.669 \pm 0.718$ | $-3.823 \pm 0.621$ | $-4.090 \pm 0.505$ |

---

**Algorithm 1** ACE Likelihood Evaluation

---

1: **Input:** $\mathbf{x}_o$, $\mathbf{x}_u$, $\mathbf{b}$
2: Set $\mathbf{x}_{cur} = \phi(\mathbf{x}_o; \mathbf{b})$ and $\mathbf{b}_{cur} = \mathbf{b}$
3: Initialize $r = 0$
4: Choose an arbitrary permutation $u'$ of $u$
5: **for** $u'_i$ **in** $u'$ **do**
6:     Compute $\log p(x_{u'_i} \mid \mathbf{x}_{cur})$ using Equation 8
7:     Set $r = r + \log p(x_{u'_i} \mid \mathbf{x}_{cur})$
8:     Set $\mathbf{x}_{cur}[u'_i] = x_{u'_i}$
9:     Set $\mathbf{b}_{cur}[u'_i] = 1$
10: **end for**
11: **Output:** $r$, which contains $\log p(\mathbf{x}_u \mid \mathbf{x}_o)$

---

---

**Algorithm 2** ACE Proposal Sampling

---

1: **Input:** $\mathbf{x}_o$, $\mathbf{b}$, $u$
2: Set $\mathbf{x}_{cur} = \phi(\mathbf{x}_o; \mathbf{b})$ and $\mathbf{b}_{cur} = \mathbf{b}$
3: Choose an arbitrary permutation $u'$ of $u$
4: **for** $u'_i$ **in** $u'$ **do**
5:     Sample $x_{u'_i} \sim q(x_{u'_i} \mid \mathbf{x}_{cur}; \mathbf{b}_{cur})$
6:     Set $\mathbf{x}_{cur}[u'_i] = x_{u'_i}$
7:     Set $\mathbf{b}_{cur}[u'_i] = 1$
8: **end for**
9: **Output:** $\mathbf{x}_{cur}$, which contains the observed and imputed values

---

---

**Algorithm 3** ACE Energy Sampling

---

1: **Input:** $\mathbf{x}_o$, $\mathbf{b}$, $u$, $N$
2: Set $\mathbf{x}_{cur} = \phi(\mathbf{x}_o; \mathbf{b})$ and $\mathbf{b}_{cur} = \mathbf{b}$
3: Choose an arbitrary permutation $u'$ of $u$
4: **for** $u'_i$ **in** $u'$ **do**
5:     Draw samples $\{x_{u'_i}^{(s)}\}_{s=1}^N$ from $q(x_{u'_i} \mid \mathbf{x}_{cur}; \mathbf{b}_{cur})$
6:     Compute importance weights for the $N$ samples, as in Equation 6
7:     Draw $x_{u'_i}$ from the $N$ samples according to the importance weights
8:     Set $\mathbf{x}_{cur}[u'_i] = x_{u'_i}$
9:     Set $\mathbf{b}_{cur}[u'_i] = 1$
10: **end for**
11: **Output:** $\mathbf{x}_{cur}$, which contains the observed and imputed values

---

Table 6: UCI datasets used in our experiments.

| Dataset | Instances | Dimensions |
|---|---|---|
| POWER | 1.66M | 6 |
| GAS | 852K | 8 |
| HEPMASS | 315K | 21 |
| MINIBOONE | 29.6K | 43 |
| BSDS | 1M | 63 |

## D  Experimental and Implementation Details

We used a fully-connected residual architecture for both the proposal and energy networks. Each network uses pre-activation residual blocks [13] and ReLU activations.

While the energy network only outputs one energy at a time, we can compute energies for every unobserved dimension in parallel by processing them as a batch. A softplus activation is applied to the network's output to ensure energies are nonnegative. We also enforce an upper bound on the energies by manually clipping the network outputs. This is equivalent to setting a lower bound on the unnormalized likelihoods, and we found it improved stability during training. A bound of 30 worked well in our experiments.

During training, we approximate normalizing constants with 20 importance samples from the proposal distribution. Proposal distributions used 10 mixture components, and the minimum allowed scale of each component was 0.001. A small amount of Gaussian noise was added to continuous values in each batch of data during training, as we found it improved stability. The learning rate was linearly annealed over the course of training. We used a warm-up period at the beginning of training where only the proposal network is optimized so that importance sampling does not occur until the proposal is sufficiently similar to the target distribution. Table 7 gives the hyperparameters that varied between datasets. Evaluations were performed using the weights that produced the highest likelihoods on a set of validation data during training.

All models except for MNIST were trained on two Tesla V100 GPUs, and training time varied from roughly a few hours to a day depending on the dataset. The MNIST model trained on four Tesla V100 GPUs for between one and two days. However, we note that multiple GPUs are not necessary for good results — we found that state-of-the-art performance can still be achieved by training ACE models on a single GPU with smaller batch sizes.

### D.1  MNIST

When training on MNIST, images were scaled to the range $[0, 1]$, and the reported likelihoods are evaluated in that space.

For MNIST, we use a different masking scheme during training so that the model learns to inpaint specific types of regions, such as square cutouts. The mask for each example is sampled from a mixture of the following distributions:

- **Bernoulli:** Each pixel is randomly selected to be observed with probability $p = 0.5$.
- **Half:** The upper, lower, left, or right half of the image is randomly selected to be observed.
- **Rectangular:** A random rectangle within the image is selected to be unobserved, with the constraint that the area of the rectangle is at least 30% of the image.
- **Square:** A square with a fourth of the area of the image is randomly selected to be unobserved.

During training (but not at test time), each sampled mask was also overlaid with an additional Bernoulli mask for $p \sim \mathcal{U}(0.02, 0.98)$ in order to help simulate the distribution of masks that the model will encounter during the autoregressive procedures it uses during inference. At test time, the extra Bernoulli noise was not used when sampling masks.

ACFlow was trained analogously to ACE, using the authors' code.

## E  Results

Table 8 presents the full UCI likelihood results with standard deviations. In the main text, the imputation results are presented as a graph. We give the values that generated the graph, along with standard deviations, in Table 9.

Table 7: Dataset-specific hyperparameters.

| HYPERPARAMETER | POWER | GAS | HEPMASS | MINIBOONE | BSDS | ADULT | MNIST |
|---|---|---|---|---|---|---|---|
| Dropout | 0.2 | 0.0 | 0.2 | 0.5 | 0.2 | 0.5 | 0.2 |
| MSE Penalty Coef. | 1.0 | 0.0 | 0.0 | 0.0 | 0.0 | 1.0 | 0.0 |
| Training Steps | 1600000 | 1000000 | 1000000 | 15000 | 1000000 | 40000 | 800000 |
| Warm-up Steps | 5000 | 5000 | 5000 | 100 | 5000 | 2500 | 100000 |
| Training Noise Scale | 0.003 | 0.001 | 0.001 | 0.005 | 0.001 | 0.005 | 0.01 |
| Learning Rate | 0.0001 | 0.001 | 0.0005 | 0.001 | 0.001 | 0.0005 | 0.0002 |
| Batch Size | 512 | 2048 | 2048 | 2048 | 2048 | 1024 | 64 |
| Proposal Hidden Dim. | 512 | 512 | 512 | 512 | 1024 | 512 | 1024 |
| Proposal Res. Blocks | 4 | 4 | 4 | 4 | 4 | 4 | 5 |
| Proposal Latent Output Dim. | 64 | 64 | 64 | 64 | 64 | 64 | 128 |



Figure 5: We use a bitmask $\mathbf{b}$ and zero-imputing function $\phi(\cdot; \mathbf{b})$ to ensure network inputs always have the same shape, regardless of how many features are observed or unobserved. In the figure, shaded cells correspond to observed features.

Table 8: Arbitrary conditional log-likelihood results for UCI datasets. Standard deviation is over 5 trials with different observed masks.

| Dataset | Method | Missing Rate | LL Mean | LL Std. |
|---|---|---|---|---|
| Power | ACE | 0.0 | 0.631 | 0.002 |
| Power | ACE Proposal | 0.0 | 0.583 | 0.003 |
| Power | ACFlow | 0.0 | 0.561 | 0.003 |
| Power | ACFlow+BG | 0.0 | 0.528 | 0.003 |
| Power | VAEAC | 0.0 | -0.042 | 0.002 |
| Power | ACE | 0.1 | 0.633 | 0.003 |
| Power | ACE Proposal | 0.1 | 0.573 | 0.003 |
| Power | ACFlow | 0.1 | 0.557 | 0.003 |
| Power | ACFlow+BG | 0.1 | 0.510 | 0.003 |
| Power | VAEAC | 0.1 | -0.103 | 0.005 |
| Power | ACE | 0.5 | 0.600 | 0.003 |
| Power | ACE Proposal | 0.5 | 0.542 | 0.003 |
| Power | ACFlow | 0.5 | 0.458 | 0.005 |
| Power | ACFlow+BG | 0.5 | 0.417 | 0.005 |
| Power | VAEAC | 0.5 | -0.343 | 0.004 |
| Gas | ACE | 0.0 | 9.643 | 0.005 |
| Gas | ACE Proposal | 0.0 | 9.484 | 0.005 |
| Gas | ACFlow | 0.0 | 8.086 | 0.010 |
| Gas | ACFlow+BG | 0.0 | 7.593 | 0.011 |
| Gas | VAEAC | 0.0 | 2.418 | 0.006 |
| Gas | ACE | 0.1 | 9.526 | 0.007 |
| Gas | ACE Proposal | 0.1 | 9.348 | 0.007 |
| Gas | ACFlow | 0.1 | 7.568 | 0.005 |
| Gas | ACFlow+BG | 0.1 | 7.212 | 0.008 |
| Gas | VAEAC | 0.1 | 2.823 | 0.009 |
| Gas | ACE | 0.5 | 8.530 | 0.007 |
| Gas | ACE Proposal | 0.5 | 8.183 | 0.005 |
| Gas | ACFlow | 0.5 | 5.405 | 0.008 |
| Gas | ACFlow+BG | 0.5 | 4.818 | 0.009 |
| Gas | VAEAC | 0.5 | 1.952 | 0.023 |
| Hepmass | ACE | 0.0 | -3.859 | 0.005 |
| Hepmass | ACE Proposal | 0.0 | -4.417 | 0.005 |
| Hepmass | ACFlow | 0.0 | -8.197 | 0.008 |
| Hepmass | ACFlow+BG | 0.0 | -6.833 | 0.006 |
| Hepmass | VAEAC | 0.0 | -10.082 | 0.010 |
| Hepmass | ACE | 0.1 | -4.255 | 0.003 |
| Hepmass | ACE Proposal | 0.1 | -4.796 | 0.003 |
| Hepmass | ACFlow | 0.1 | -7.784 | 0.006 |
| Hepmass | ACFlow+BG | 0.1 | -9.670 | 0.007 |
| Hepmass | VAEAC | 0.1 | -10.389 | 0.005 |
| Hepmass | ACE | 0.5 | -8.133 | 0.007 |
| Hepmass | ACE Proposal | 0.5 | -8.497 | 0.006 |
| Hepmass | ACFlow | 0.5 | -10.538 | 0.006 |
| Hepmass | ACFlow+BG | 0.5 | -10.975 | 0.006 |
| Hepmass | VAEAC | 0.5 | -11.415 | 0.012 |
| Miniboone | ACE | 0.0 | 0.310 | 0.054 |
| Miniboone | ACE Proposal | 0.0 | -0.241 | 0.056 |
| Miniboone | ACFlow | 0.0 | -0.972 | 0.022 |
| Miniboone | ACFlow+BG | 0.0 | -1.098 | 0.032 |
| Miniboone | VAEAC | 0.0 | -3.452 | 0.067 |
| Miniboone | ACE | 0.1 | -0.688 | 0.046 |
| Miniboone | ACE Proposal | 0.1 | -1.328 | 0.057 |
| Miniboone | ACFlow | 0.1 | -5.150 | 0.053 |
| Miniboone | ACFlow+BG | 0.1 | -3.577 | 0.057 |
| Miniboone | VAEAC | 0.1 | -4.242 | 0.071 |
| Miniboone | ACE | 0.5 | -5.701 | 0.050 |
| Miniboone | ACE Proposal | 0.5 | -9.169 | 0.083 |
| Miniboone | ACFlow | 0.5 | -9.892 | 0.084 |
| Miniboone | ACFlow+BG | 0.5 | -10.849 | 0.105 |
| Miniboone | VAEAC | 0.5 | -9.051 | 0.079 |
| BSDS | ACE | 0.0 | 86.701 | 0.008 |
| BSDS | ACE Proposal | 0.0 | 85.228 | 0.009 |
| BSDS | ACFlow | 0.0 | 81.827 | 0.007 |
| BSDS | ACFlow+BG | 0.0 | 81.399 | 0.008 |
| BSDS | VAEAC | 0.0 | 74.850 | 0.005 |
| BSDS | ACE | 0.1 | 86.130 | 0.022 |
| BSDS | ACE Proposal | 0.1 | 84.204 | 0.020 |
| BSDS | ACFlow | 0.1 | 80.783 | 0.018 |
| BSDS | ACFlow+BG | 0.1 | 79.745 | 0.017 |
| BSDS | VAEAC | 0.1 | 74.313 | 0.015 |
| BSDS | ACE | 0.5 | 80.613 | 0.027 |
| BSDS | ACE Proposal | 0.5 | 75.767 | 0.131 |
| BSDS | ACFlow | 0.5 | 75.050 | 0.010 |
| BSDS | ACFlow+BG | 0.5 | 73.061 | 0.015 |
| BSDS | VAEAC | 0.5 | 66.628 | 0.029 |

Table 9: Imputation results for UCI datasets. Standard deviation is over 5 trials with different observed masks.

| Dataset | Method | Missing Rate | NRMSE Mean | NRMSE Std. |
|---|---|---|---|---|
| Power | ACE | 0.0 | 0.828 | 0.002 |
| Power | ACE Proposal | 0.0 | 0.828 | 0.002 |
| Power | ACFlow | 0.0 | 0.877 | 0.001 |
| Power | ACFlow+BG | 0.0 | 0.833 | 0.002 |
| Power | VAEAC | 0.0 | 0.880 | 0.001 |
| Power | ACE | 0.1 | 0.653 | 0.000 |
| Power | ACE Proposal | 0.1 | 0.653 | 0.000 |
| Power | ACFlow | 0.1 | 0.877 | 0.002 |
| Power | ACFlow+BG | 0.1 | 0.836 | 0.002 |
| Power | VAEAC | 0.1 | 0.881 | 0.003 |
| Power | ACE | 0.5 | 0.831 | 0.000 |
| Power | ACE Proposal | 0.5 | 0.831 | 0.000 |
| Power | ACFlow | 0.5 | 0.890 | 0.000 |
| Power | ACFlow+BG | 0.5 | 0.843 | 0.001 |
| Power | VAEAC | 0.5 | 0.892 | 0.002 |
| Gas | ACE | 0.0 | 0.335 | 0.027 |
| Gas | ACE Proposal | 0.0 | 0.312 | 0.033 |
| Gas | ACFlow | 0.0 | 0.567 | 0.050 |
| Gas | ACFlow+BG | 0.0 | 0.369 | 0.016 |
| Gas | VAEAC | 0.0 | 0.574 | 0.033 |
| Gas | ACE | 0.1 | 0.135 | 0.014 |
| Gas | ACE Proposal | 0.1 | 0.077 | 0.000 |
| Gas | ACFlow | 0.1 | 0.588 | 0.025 |
| Gas | ACFlow+BG | 0.1 | 0.384 | 0.018 |
| Gas | VAEAC | 0.1 | 0.558 | 0.047 |
| Gas | ACE | 0.5 | 0.404 | 0.052 |
| Gas | ACE Proposal | 0.5 | 0.325 | 0.000 |
| Gas | ACFlow | 0.5 | 0.488 | 0.030 |
| Gas | ACFlow+BG | 0.5 | 0.421 | 0.016 |
| Gas | VAEAC | 0.5 | 0.531 | 0.036 |
| Hepmass | ACE | 0.0 | 0.830 | 0.001 |
| Hepmass | ACE Proposal | 0.0 | 0.832 | 0.001 |
| Hepmass | ACFlow | 0.0 | 0.909 | 0.000 |
| Hepmass | ACFlow+BG | 0.0 | 0.861 | 0.001 |
| Hepmass | VAEAC | 0.0 | 0.896 | 0.001 |
| Hepmass | ACE | 0.1 | 0.610 | 0.000 |
| Hepmass | ACE Proposal | 0.1 | 0.623 | 0.000 |
| Hepmass | ACFlow | 0.1 | 0.908 | 0.001 |
| Hepmass | ACFlow+BG | 0.1 | 0.863 | 0.001 |
| Hepmass | VAEAC | 0.1 | 0.899 | 0.000 |
| Hepmass | ACE | 0.5 | 0.858 | 0.000 |
| Hepmass | ACE Proposal | 0.5 | 0.858 | 0.000 |
| Hepmass | ACFlow | 0.5 | 0.938 | 0.000 |
| Hepmass | ACFlow+BG | 0.5 | 0.890 | 0.000 |
| Hepmass | VAEAC | 0.5 | 0.915 | 0.001 |
| Miniboone | ACE | 0.0 | 0.432 | 0.003 |
| Miniboone | ACE Proposal | 0.0 | 0.436 | 0.004 |
| Miniboone | ACFlow | 0.0 | 0.478 | 0.004 |
| Miniboone | ACFlow+BG | 0.0 | 0.442 | 0.001 |
| Miniboone | VAEAC | 0.0 | 0.462 | 0.002 |
| Miniboone | ACE | 0.1 | 0.346 | 0.001 |
| Miniboone | ACE Proposal | 0.1 | 0.355 | 0.000 |
| Miniboone | ACFlow | 0.1 | 0.533 | 0.005 |
| Miniboone | ACFlow+BG | 0.1 | 0.468 | 0.003 |
| Miniboone | VAEAC | 0.1 | 0.467 | 0.004 |
| Miniboone | ACE | 0.5 | 0.497 | 0.000 |
| Miniboone | ACE Proposal | 0.5 | 0.500 | 0.000 |
| Miniboone | ACFlow | 0.5 | 0.614 | 0.004 |
| Miniboone | ACFlow+BG | 0.5 | 0.582 | 0.007 |
| Miniboone | VAEAC | 0.5 | 0.513 | 0.004 |
| BSDS | ACE | 0.0 | 0.525 | 0.000 |
| BSDS | ACE Proposal | 0.0 | 0.535 | 0.000 |
| BSDS | ACFlow | 0.0 | 0.603 | 0.000 |
| BSDS | ACFlow+BG | 0.0 | 0.572 | 0.000 |
| BSDS | VAEAC | 0.0 | 0.615 | 0.000 |
| BSDS | ACE | 0.1 | 0.389 | 0.000 |
| BSDS | ACE Proposal | 0.1 | 0.407 | 0.000 |
| BSDS | ACFlow | 0.1 | 0.610 | 0.000 |
| BSDS | ACFlow+BG | 0.1 | 0.586 | 0.001 |
| BSDS | VAEAC | 0.1 | 0.620 | 0.000 |
| BSDS | ACE | 0.5 | 0.560 | 0.000 |
| BSDS | ACE Proposal | 0.5 | 0.579 | 0.000 |
| BSDS | ACFlow | 0.5 | 0.667 | 0.001 |
| BSDS | ACFlow+BG | 0.5 | 0.645 | 0.000 |
| BSDS | VAEAC | 0.5 | 0.666 | 0.001 |