# OpenReview forum: "Arbitrary Conditional Distributions with Energy"
_NeurIPS.cc/2021/Conference — NeurIPS 2021 Poster_

### Official Review · Reviewer_BNRH · 2021-07-09

**Rating:** 6
**Confidence:** 4

**Summary:**

Given D random variables, the number of possible conditional distributions is exponential in D. The authors propose a method for efficiently estimating this exponentially large set of distributions with a compact autoregressive energy-based model. The method is applied to conditional density estimation and imputation tasks on UCI repository datasets and MNIST, showing favourable performance compared to strong baselines.

**Limitations And Societal Impact:**

The main potential limitation I see is that the method may be more computationally costly than the baselines, and I couldn't see a discussion of this in the main text.

Otherwise, I thought the authors were honest about any limitations, and I agree that there are no obvious negative societal impacts from their work.

**Main Review:**

**Overall verdict**

This is an extremely well-written paper with decent empirical results on an important task. The major drawback is the lack of methodological novelty; readers who have read the prior literature may see this paper as somewhat 'obvious'. Overall, I still tend towards acceptance, so long as the authors alter section 4.2/4.3 to *explicitly* state that the contents are adapted from Nash & Durkan (2019).


**Originality**

Methodologically, this paper is not especially novel. The proposed method--ACE--is a relatively straightforward combination of two existing ideas: 1) The use of a single autoregressive model to model all possible conditionals (Uria et al., 2014) and 2) The use of an energy function to specify the univariate conditionals of this autoregressive model (Nash & Durkan, 2019).

However, the paper is more original in the task that is solves. Prior work has largely been concerned with joint density estimation, whereas this paper targets conditional density estimation. I say 'largely' because the methods of Uria et al. and Germain et al. (2015) can clearly be used for conditional density-estimation, but their experiments are not focused on this.

In a sense then, this paper does two things:

1) Reminds the community of an overlooked fact, namely that a single autoregressive model can be used for arbitrary conditional density estimation
2) Demonstrates that such an approach, when combined with modern energy-based architectures, achieves decent empirical performance


**Quality**

Since the paper lacks significant methodological or theoretical contributions, its technical quality largely depends on the experiments.

I think the experiments are of an *acceptable* quality. The authors follow a fairly standard setup of UCI + MNIST datasets and assess conditional log-likelihoods + RMSE of imputations. The proposed method clearly and consistently outperforms a strong baseline of ACFlow. Moreover, its very nice that ACE is applicable to continuous & discrete data.

Could the authors comment on the relative computational cost of ACE versus ACFlow? Both in terms of training costs (iterations & wallclock time until convergence) and cost of inference/sampling?


**Relation to prior work**

In various places, the authors cite Nash & Durkan (2019) who introduced Autoregressive Energy Machines (AEM). However, there is insufficient explanation of how AEM and ACE are related. This is highly problematic, as most of Sections 4.2 and 4.3 are adapted from Nash & Durkan without sufficiently explicit attribution. Specifically, Nash & Durkan introduced the idea of jointly training an autoregressive energy model and proposal using dimension-wise importance sampling.

One key difference between AEM and ACE is the way in which they ensure that autoregressive constraints are satisfied. AEM uses autoregressive-preserving masking in the hidden layers of the network, whilst ACE avoids this by duplicating the input (one copy for each unobserved variable) and applying different masks to each copy. Both are valid strategies, but the former is more computationally efficient, raising the question of why the authors chose the second strategy. In fact, the ACE-style strategy was introduced very early on in NADE (Larochelle & Murray, 2011), and was later made more computationally efficient with hidden-layer masking in MADE (Germain et al., 2015). Arguably, therefore, this paper is taking a step backwards.

**Clarity**

The paper is exceedingly well-written.

**Significance**

I think this paper is of moderate significance to the generative-modelling-with-missing-data literature. Autoregressive models have generally been overlooked in this literature, despite the early contributions by Uria et al. (2014).


**References**

Uria, Benigno, Iain Murray, and Hugo Larochelle. "A deep and tractable density estimator." International Conference on Machine Learning. PMLR, 2014.

Larochelle,  Hugo  and  Murray,  Iain.   The  neural  autoregressive distribution estimator. In Proceedings of the14th International Conference on Artificial Intelligenceand Statistics (AISTATS 2011), volume 15, pp. 29–37, Ft.Lauderdale, USA, 2011. JMLR W&CP.

M. Germain, K. Gregor, I. Murray, and H. Larochelle.  MADE: Masked Autoencoder for DistributionEstimation.Proceedings of the 32nd International Conference on Machine Learning, pages 881–889,2015

Nash, C. & Durkan, C.. (2019). Autoregressive Energy Machines. Proceedings of the 36th International Conference on Machine Learning, in Proceedings of Machine Learning Research 97:1735-1744 Available from http://proceedings.mlr.press/v97/durkan19a.html.


**Time Spent Reviewing:**

4

---

> ### Author Response · Authors · 2021-08-10
> **Author's Response**
>
> Thank you for your thoughtful suggestions and questions. We shall more explicitly cite Nash & Durkan (2019) for introducing the idea of jointly training an autoregressive energy model and proposal distribution using dimension-wise importance sampling. We choose not to use autoregressive-preserving masking, like AEMs, because during training we don’t actually compute autoregressive likelihoods -- we only care to optimize 1d conditionals. By enforcing a masking constraint on the network, AEMs effectively mandate a single autoregressive ordering. This would prevent ACE from being able to evaluate likelihoods in any order.
>
> In terms of training cost, ACE performs similarly to ACFlow in most cases. Both methods can be trained in anywhere from a few hours to 1-2 days (on a single GPU), depending on the dataset. When sampling $p(x_u | x_o)$ after training, in general, sampling is computationally similar in both methods. In the special case when ACFlow has an independent base distribution for the change of variables applied to $x_u$, it bypasses an autoregressive procedure, which may be more efficient. When evaluating likelihoods, ACE can compute the autoregressive components in parallel (unlike an autoregressive RNN in ACFlow).

---

> > ### Comment · Reviewer_BNRH · 2021-09-11
> > **Thank you for your response**
> >
> > Thank you for your comments. My opinion of the paper remains the same.
> >
> > I would like to re-emphasise that explaining the contributions of Nash & Durkan is vital; multiple reviewers here were unaware of their contributions after reading your paper.
> >
> > > By enforcing a masking constraint on the network, AEMs effectively mandate a single autoregressive ordering
> >
> > Whilst a single mask does mandate a single ordering, you can sample a different mask for each minibatch. See section 4.2 of Germain et al (2015) on 'order-agnostic training'. The paper should be amended to make it clear that this is a valid alternative.

---

### Official Review · Reviewer_4qhh · 2021-07-12

**Rating:** 6
**Confidence:** 3

**Summary:**

The main contribution of this paper is a new method for data imputation. The presented method is simpler than existing ones, and shows good results in conditional likelihood estimation and in some cases in data imputation. However, experimental results in data imputation are difficult to compare to existing ones, and the authors use a smaller number of datasets to evaluate their method, compared to preceding works.

**Limitations And Societal Impact:**

Main limitation: lack of significant experiments in imputation (see Main Review).

**Main Review:**

## Detailed summary

Given a dataset $D$, this article presents a method for:
1) estimating the conditional likelihood of a subset A of the coordinates of a vector $x \in D$, given a subset B disjoint from A. A and B can be chosen arbitrarily;
2) imputing a missing part of a vector $x \in D$.

The proposed method is both simpler and more accurate than the preceding ones.

The proposed method is broadly based on the paper *Autoregressive energy machines*, Nash 2019. It consists in training jointly two neural networks:
1) a proposal network, which outputs a conditional distribution of unobserved coordinates $x_u$ of $x$, given observed ones $x_o$; it also outputs a latent vector $\gamma$;
2) an energy network, which takes as input $x_o$, the latent vector $\gamma$, and a proposition $x_{u_i}$ for the $i$-th coordinate of $x_u$, and outputs the *energy* associated to such a configuration.

After training, each of these networks can be used independently: the proposal network can be use to impute missing data, the energy network can be used to measure the likelihood of the coordinates of a vector given the others.

## Experiments

The authors test their method with two preceding ones: VAEAC and ACFlow. Their method performs better. The choice of datasets is the same as in the paper introducing ACFlow.

According to the authors, this restricted choice of concurrent methods is due to the fact that GAN methods (GAIN and MisGAIN) are known to perform poorly compared to VAEAC [1].

However, about imputation:
1) the tested datasets in [1] are not tested here (except MNIST with mask);
2) the experiment made on MNIST in [1] is not comparable to the one presented in this paper: in Fig. 3a, the hidden part is much smaller than in [1], Fig. 1; anyway, there is no comparison with VAEAC in this setup;
3) the same remark holds when comparing to ACFlow [2]: how can we be sure that results in Fig. 1 are representative of the results obtained with ACE? It would be fair to compare ACE on the same digit, partly hidden with the same mask as in [1] and [2]. This way, the reader would be sure than the presented results are fair;
4) contrary to [1] and [2], there is no experiment with CelebA and Omniglot;
5) GAN methods are assumed to perform poorly on the chosen datasets, but there is no evidence of that (the datasets used here differ from [1], in which such experiments have been made).

## Clarity

Overall, the paper is well written.

## Originality

The proposed method is an adaptation of an existing one. However, the application to arbitrary conditioning is novel.

## Significance

Data imputation is a major question in AI.

The proposed method is simple, which is the main advantage of the proposed method.

Although experimental results are encouraging when estimating the likelihood of vectors, the only clear result in imputation is in Fig. 2, which insufficient.

## References

[1] *Variational Autoencoder with Arbitrary Conditioning*, Ivanov 2019

[2] *ACFlow: Flow Models for Arbitrary Conditional Likelihoods*, Li 2019

## Rebuttal

I have read the authors' rebuttal, and I have several remarks:
 * I understand the issue raised by the authors about image completion, especially for medium size images, such as in CelebA or Omniglot;
 * the supplementary experiments made by the authors are encouraging, and may be put in the final paper (or at least in the Appendix), since they challenge [1] on 3 of the 5 datasets presented in Table 1 of [1]. However, I am curious to see a comparison with [1] on the 2 remaining datasets (Zoo and Phishing).

With these 3 new experiments, the evidence that the presented method works better than VAEAC (and ACFlow?) is stronger. However, to be completely fair, a comparison should be made with the two remaining datasets. I have checked out Ivanov's code, and I confirm that the code related to these datasets is missing, so the 3 tested datasets have not been "cherry-picked".

I tend to be mostly satisfied with the authors' answer. Weak accept.

**Time Spent Reviewing:**

7

---

> ### Author Response · Authors · 2021-08-09
> **Author's Response**
>
> Thank you for your review and helpful comments and suggestions. We note that the focus of our work is the study of arbitrary conditioning on general tabular data (without any specialized image domain architectures, such as CNNs). As such, we used the POWER, GAS, HEPMASS, MINIBOONE, and BSDS300 datasets, which are are standard tabular benchmark datasets for the study of likelihoods (e.g. Masked Autoregressive Flow for Density Estimation, NeurIPS 2017; Transformation Autoregressive Networks, ICML 2018; Neural Spline Flows, NeurIPS 2019, etc.). In addition, we tested higher dimensional estimation via a flattened (e.g. vectorized) study of MNIST. Here, we tested a tabular treatment of pixel-values without any specialized architectures with the same framework as used in general data. To test performance as the number of features increased, we held the architectures across models to remain the same as in the general tabular data case. We extended further than existing arbitrary conditional experimentation by performing an additional experiment in the mixed continuous/discrete case with the UCI Adult dataset (section 5.3), which was not considered by Ivanov et al (2019) or Li et al. (2019). For completeness, we have now also run our method on the three tabular datasets that Ivanov et al (2019) provide code for: WhiteWine, Yeast, Mushroom. Here, we see that even with minimal hyperparameter tuning, we obtain better imputation results (see table below, which is analogous to Table 1 in Ivanov et al (2019)). The aggregate of these experiments yields a thorough evaluation of our methods’ ability to perform arbitrary conditioning in general tabular data, and shows that ACE is a state-of-the-art method for estimating arbitrary likelihoods. Thus, we believe that this work is of interest to the ML community and hope that you recommend our paper for publication.
>
> |      | White Wine | Yeast | Mushroom
> | ----------- | ----------- |  ----------- |  ----------- |
> | VAEAC      | 0.850 ± 0.007 | 0.94 ± 0.01 | 0.244 ± 0.002 |
> | ACE Proposal   | **0.842 ± 0.003** | **0.91 ± 0.03** | **0.232 ± 0.001**|
> | ACE | 0.853 ± 0.006 | 0.92 ± 0.02 | **0.232 ± 0.001**|
>
> We also note that Li et al provide results that show that GAIN, as well as some other common imputation methods, don’t perform as well as ACFlow on the imputation tasks that we consider (and thereby don’t perform as well as ACE). In order to make this additional context clear and more accessible in our paper, we will happily include those numbers directly.

---

### Official Review · Reviewer_qrbW · 2021-07-16

**Rating:** 6
**Confidence:** 4

**Summary:**

This paper proposes arbitrary conditioning with energy (ACE), a method for density estimation that can access any conditional distribution over any subset of random variables. It presents learning and inference procedures for ACE, and validates ACE's performance on arbitrary conditional/marginal density estimation and data imputation for real-valued, high-dimensional and mixed continuous-discrete data.

**Limitations And Societal Impact:**

Yes

**Main Review:**

Originality: the method is new, but its claim of novelty is misleading. The paper claims that it is the first work to leverage the reduction of arbitrary conditioning to 1d conditional estimations. This claim seems too strong. For example autoregressive methods like [31] can similarly handle arbitrary conditioning, as long as the variables to be conditioned on are at the beginning and the variables to be marginalized are at the end. Similar points are made in *Query Training: Learning a Worse Model to Infer Better Marginals in Undirected Graphical Models with Hidden Variables, Miguel Lázaro-Gredilla, Wolfgang Lehrach, Nishad Gothoskar, Guangyao Zhou, Antoine Dedieu, Dileep George, AAAI 2021* (a missing citation that explores very similar ideas and should be covered at least in related works), section *Queries that Need to Be Answered*. It is true that none of these works pursue this task, but the way the paper is framing the contributions should be weakened. To me the main contribution seems to be the use of an energy-based formulation together with an importance sampling based approach, which gives the model useful flexibility to outperform pervious methods based on parametric distributions.

Quality and clarify: the paper is technically sound, and clearly written for the most part. There are some points that need clarifying:
1. Equation 8 is quite confusing. How are the x_o and x_u partitioned? Where is the autoregressive component?
2. For likelihood evaluations in 4.4.1, it's not clear how ordering is handled. Is the method essentially computing the product of all possible 1d conditional distributions? If so, even though these can be computed in parallel, there are still a huge number of conditional distributions and it looks very expensive. If the method is using a particular order, then different orders would give different results and it's not clear how the numbers in table 1 are derived.
3. For ACE with proposal, since it's using a mixture of Gaussians (i.e. a parametric distribution), it looks similar in idea to [31]. While it's good to see that having the additional flexibility from the energy function gives further performance boost, I'm also curious to see what a method like [31] would perform.

Significance: the studied problem is an interesting and important problem, and the proposed ACE method can potentially be very useful.

One other question: ACE uses importance sampling. How many samples are used? Would the Monte Carlo estimates result in high variance, e.g. in gradients estimations? How does that affect the learning dynamics?

===================================
Update after rebuttal:

During the discussion period it is brought to my attention that the use of an energy-based formulation in combination with importance sampling is also not new. In that case the methodology seems like a (somewhat trivial) combination of several existing pieces, and I ask the authors to weaken their claims of novelty. The experimental results are still interesting and relevant, so I'm maintaining my current score (with the assumption that claims of novelty will be weakened).


**Time Spent Reviewing:**

3

---

> ### Author Response · Authors · 2021-08-09
> **Author's Response**
>
> Thank you for your thoughtful questions and suggestions. Thank you for the pointer to _Query Training_, we will gladly include it in the related works. Our emphasis and contribution is the advancement of arbitrary conditioning on general tabular data through 1d conditionals and how to flexibly estimate 1d conditionals for this task. We note that our model is actually not trained autoregressively, instead it is trained directly on 1d conditionals. Eq. 8 lists the 1d conditionals that are possible for a given $x_o$, where the subset of features $o$, is chosen uniformly at random. A single order for evaluating likelihoods is chosen at random. The overall variance among orderings is relatively low. Please see our discussion in Appendix C for further details. We report the results of directly using the parameters of the GMMs as “ACE Proposal” in the experiments section.

---

### Official Review · Reviewer_LU53 · 2021-07-23

**Rating:** 6
**Confidence:** 4

**Summary:**

This paper proposes a novel technique (ACE) for training a density estimator that allows for arbitrary conditioning and marginalization.  The core idea is to decompose the target conditional distribution into a product of autoregressive 1-D conditionals.  Each scalar conditionals are substantially easier to model than their joint counterparts.  The training scheme consists of two components, denoted proposal and energy networks.  Energy network is used to model the energy function that models the target distribution, and the proposal network is to aid in the important sampling procedure used to estimate the normalizing constant for the energy function.  The paper has convincing and thorough experimental results that compare against several representative baselines, where ACE exhibits a strong performance gain over them.

**Limitations And Societal Impact:**

The authors briefly discuss potential societal impact of their work.

**Main Review:**

### Strengths
- The paper presents a simple idea that is shown to be effective experimentally.  The technique is very sensible, and it overcomes several disadvantages that existing methods have.
- Writing is very clear and easy to follow.

### Concerns
- Estimating marginals: The paper claims that ACE can be used to estimate arbitrary marginal distributions, but the paper doesn't explicitly state how it can be done.  How is this done?  One very naive way to compute the marginal $p(x_o)$ would be to fix some random values for the remaining variables ($x_u$), then computing $p(x_o) = p(x_u, x_o) / p(x_u \vert x_o)$.  I'm concerned that this marginal density estimator likely has very high variance because. (1) Both $p(x_u, x_o)$ and $p(x_u \vert x_o)$ suffer from high variance due to model inaccuracy and dependence on the ordering, and (2) The choice of $x_u$ presumably affects both of these quantities heavily.  This is seemingly a much more task than estimating conditional distributions, so it'd be important to include experimental results for marginal density estimation.  Also, additional questions include:
  - Is this naive method how one would estimate an arbitrary marginal, or is there a better way?
  - If the naive method is what one should use, how does ACE perform, and how much variance is there?

- Justification for the loss function: The loss function, while reasonable, feels a bit arbitrary.  First, I am curious how important it is for the energy and proposal networks to share weights (as was done in this paper).  An interesting ablation study would be to separately train a proposal network then using it to train the energy network.  Second, how accurate is the estimation of the normalizing constant?  How many importance samples ($S$) were necessary?

- Baseline: It would have been nice to consider Locally Masked Convolution for Autoregressive Models [1] as an additional baseline, which I believe is a very relevant baseline to ACE.

### Comment
I enjoyed reading this paper, and I found the idea very simple yet effective.  Experiments and the technical motivation could be a bit stronger, but I believe that the submission is overall good.

#### References
[1] Ajay Jain, Pieter Abbeel, Deepak Pathak. Locally Masked Convolution for Autoregressive Models. In Conference on Uncertainty in Artificial Intelligence (UAI), 2020.

**Time Spent Reviewing:**

3

---

> ### Author Response · Authors · 2021-08-09
> **Author's Response**
>
> Thank you for your thoughtful review and comments. We are glad you enjoyed the paper and appreciated the simplicity of our approach. We compute marginals via a special case where we condition on nothing (empty set). That is, we compute $p(x_u) = p(X_u = x_u, | X_o = \emptyset)$, which we can compute via the product of 1d conditionals $\prod_{j=1}^{|u|} p(x_{u_j} | X_o=(x_{u_1}, \dots, x_{u_{j-1}}))$ for any permutation of $u$. For example, we could compute the marginal $p(x_7, x_2, x_3)$ as $p(x_7) p(x_2 | x_7) p(x_3 | x_7, x_2)$. Note that the 1d conditionals are estimated in the same fashion with ACE. Thus, ACE requires no extra modification to handle marginals.  We shall expand our discussion of this in the background section for additional clarity. ACE requires a proposal distribution for importance sampling. When a proposal distribution is not provided apriori, then we jointly train the proposal along with the energy network. During training we use 20 importance samples. See Table 1 for the number of samples used with each dataset during inference. Nash & Durkan (2019) (Figure 5) detail the accuracy of importance sampling in a related estimation task. We shall add a pointer to related work “Locally Masked Convolution for Autoregressive Models”; we compared to tabular (non-convolutional) approaches to study arbitrary conditioning in a general setting.

---

> > ### Comment · Reviewer_LU53 · 2021-08-27
> > **Thank you**
> >
> > Thanks for the clarification.  Conditioning on empty set makes sense.

---

### Decision · Program_Chairs · 2021-09-28

**Decision:**

Accept (Poster)

**Comment:**

The authors propose a method for estimating conditional marginal distributions under arbitrary choices of the conditioning and marginalization sets using a decomposition into single-variable conditionals (autoregressive) using a single (neural network based) energy function to model all possible conditionals.  In general, the reviewers were positive but very reserved about the work; given the existence of several previous related works, the method was viewed as incremental or even obvious to experts in the area, and felt that the experimental validation is not strong.  On the whole reviewers thought the work's ideas were generally worthy of publication, but viewed the current paper as only being right on the border for the NeurIPS conference.

**Consistency Experiment:**

NeurIPS has a long history of experimentation. In 2014, NeurIPS ran an experiment in which 10% of submissions were reviewed by two independent committees to quantify the randomness in the review process. This year, we repeated a variant of this experiment to see how the quality of the review process has changed over time.  This paper was part of the experiment and was therefore assigned to two committees (consisting of reviewers, an Area Chair, and a Senior Area Chair) that reached independent decisions.  If both committees made the same recommendation, this recommendation was followed. If a single committee recommended acceptance, the paper was accepted (with the exception of a few cases in which the other committee identified what we considered a fatal flaw, e.g., an error in a key result).

This copy’s committee reached the following decision: **Reject**

The other committee assigned to the paper recommended **Accept (Poster)**.  You can find the other set of reviews, along with any follow up discussion with the authors here:
https://openreview.net/forum?id=_idcJrecij